# Light Exposure as a Tool to Enhance the Regenerative Potential of Adipose-Derived Mesenchymal Stem/Stromal Cells

**DOI:** 10.3390/cells14151143

**Published:** 2025-07-24

**Authors:** Kaarthik Sridharan, Tawakalitu Okikiola Waheed, Susanne Staehlke, Alexander Riess, Mario Mand, Juliane Meyer, Hermann Seitz, Kirsten Peters, Olga Hahn

**Affiliations:** 1Institute of Cell Biology, Rostock University Medical Center, 18057 Rostock, Germany; kaarthikgauthum@gmail.com (K.S.); okikiolawaheed@gmail.com (T.O.W.); susanne.staehlke@med.uni-rostock.de (S.S.); kirsten.peters@med.uni-rostock.de (K.P.); 2Chair of Microfluidics, Faculty of Mechanical Engineering and Marine Technology, University of Rostock, 18059 Rostock, Germany; alexander.riess@uni-rostock.de (A.R.); mario.mand@mail.de (M.M.); hermann.seitz@uni-rostock.de (H.S.); 3Human Med AG, 19061 Schwerin, Germany; juliane.meyer@humanmed.com; 4Department of Life, Light and Matter, University of Rostock, 18059 Rostock, Germany

**Keywords:** photobiomodulation (PBM), human adipose-derived mesenchymal stem/stromal cells (adMSCs), reactive oxygen species (ROS), cytokines, migration, adipogenic differentiation

## Abstract

Photobiomodulation (PBM) utilizes different wavelengths of light to modulate cellular functions and has emerged as a promising approach in regenerative medicine. In this study, we examined the effects of blue (455 nm), red (660 nm), and near-infrared (810 nm) light, both individually and in combination, on human adipose-derived mesenchymal stem/stromal cells (adMSCs). A single, short-term exposure of adMSCs in suspension to these wavelengths using an integrating sphere revealed distinct wavelength- and dose-dependent cellular responses. Blue light exposure led to a dose-dependent increase in intracellular reactive oxygen species, accompanied by reduced cell proliferation, metabolic activity, interleukin-6/interleukin-8 secretion, and adipogenic differentiation. In contrast, red and near-infrared light preserved cell viability and metabolic function while enhancing cell migration, consistent with their documented ability to stimulate proliferation and mitochondrial activity in mesenchymal stem cells. These findings highlight the necessity of precise wavelength and dosage selection in PBM applications and support the potential of PBM as a customizable tool for optimizing patient-specific regenerative therapies.

## 1. Introduction

Light-based therapies, which utilize an approach known as photobiomodulation (PBM), have emerged as promising strategies for modulating cellular behavior and promoting tissue repair [1]. These therapies employ wavelengths of light to influence biological processes at the cellular and molecular levels [2]. The versatility of light-based approaches lies in their ability to target distinct cellular signaling pathways depending on the wavelength applied [3]. Thus, PBM can synergistically enhance cell proliferation, differentiation, and migration, regulate inflammation, and thereby improve tissue regeneration [4].

Blue light (wavelengths around 410–470 nm) has been shown to have a direct effect on mitochondrial activity by interacting with flavins and porphyrins within the electron transport chain (ETC) [5]. There is evidence that blue light can affect ETC activity, increase reactive oxygen species (ROS) production, and alter mitochondrial dynamics [6], for example, by influencing fission and fusion [7,8]. Furthermore, blue light can modulate cellular activity through the activation of opsins (e.g., melanopsin/OPN4 in mesenchymal stem cells/MSC) [9] and transient receptor potential (TRP) channels [10]. These blue light-induced effects are highly dose- and context-dependent, leading to stimulatory or inhibitory outcomes [5].

In contrast, red and near-infrared (NIR) light (wavelengths between 630 and 1000 nm) are supposed to be absorbed by mitochondrial chromophores, particularly cytochrome c oxidase (CCO)—a key enzyme in the ETC [11]. This interaction can enhance adenosine triphosphate (ATP) production through increased oxidative phosphorylation, facilitate the release of nitric oxide by photodissociation from CCO, and regulate ROS levels [11,12]. These biochemical changes are suggested to collectively support essential cellular processes such as proliferation, migration, and differentiation, which are critical for tissue regeneration [13].

Light-based therapies have gained attention in regenerative medicine due to their potential to precisely target cell functions. The therapeutic efficacy depends on several variables, including light source characteristics (e.g., wavelength and power density), energy delivery parameters, and individual differences in tissue properties such as skin thickness or pigmentation when applied topically [4,14,15,16]. The depth to which light penetrates the skin is primarily determined by its wavelength. Longer wavelengths can penetrate deeper—up to 5.4 mm—because they experience less scattering and absorption within the tissue [17,18]. In contrast, blue light interacts strongly with tissue components such as hemoglobin and melanin, resulting in a much shallower skin penetration depth, typically less than one millimeter [19,20]. Dose–response relationships further complicate the results, as the effects vary with energy density, output power, and the cell type observed [21,22,23,24,25]. Notably, the reactions to light can follow a biphasic dose–response curve, where both insufficient and excessive light doses may be suboptimal [26]. These interdependencies underscore the need for further optimization and standardization of therapeutic protocols to maximize light-based therapies’ clinical potential.

Adipose-derived mesenchymal stem/stromal cells (adMSCs) are a subpopulation of multipotent stem cells that can be isolated from adipose tissue [27]. adMSCs exhibit significant potential in regenerative medicine due to their ease of isolation, high yield compared to bone marrow-derived MSC, and ability to differentiate into various lineages, including adipogenic, chondrogenic, and osteogenic cells [28,29]. In addition, adMSCs secrete cytokines (including pro- and anti-inflammatory interleukins/ILs), growth factors, and extracellular vesicles that promote angiogenesis, modulate immune responses, and support tissue repair [30,31]. These properties make adMSCs an ideal candidate for therapeutic regenerative applications such as wound healing [31].

Recent studies have demonstrated that exposure of adMSCs to different wavelengths of light can modulate cellular functions that are critical for regenerative potential. For instance, exposure to red light at 660 nm promoted the proliferation of adMSCs [32], while both red light at 635 nm and NIR light at 809 nm enhanced osteogenic differentiation and increased mineralization in adMSCs [33]. Exposure to adMSCs with red light (630 nm) affected the release of inflammatory factors [34]. Conditioned media from adMSCs exposed to combined red light (635 and 655 nm) are shown to affect the expression of key regulators in fibrosis modulation (i.e., transforming growth factor β1 and Notch-1) [35]. These effects are supposed to be mediated by light-induced changes in mitochondrial activity, ROS levels, and the release of bioactive molecules [36]. However, these effects depend heavily on the experimental setup, particularly on the exact wavelength applied, the duration of exposure, the intensity, and the number of exposures.

The exposure of adMSCs to blue and green light (415 and 540 nm) inhibits proliferation and reduces ATP levels, indicating an impairment of cellular metabolism [37]. Bone marrow-derived MSCs exposed to blue light (470 nm) showed reduced proliferation, reduced osteogenic differentiation, and apoptosis, possibly due to ROS accumulation and DNA damage. However, direct adMSCs-specific data on blue light exposure are limited.

In this in vitro study, we exposed pre-cultivated adMSCs representing a subpopulation of adipose tissue-derived cells with a high regenerative capacity to light of different wavelengths using an unconventional approach designed to simulate a clinical scenario. To ensure uniform illumination, we used an integrating sphere (‘Ulbricht sphere’), which diffuses the incoming light multiple times on its inner surface, thereby providing homogenous exposure to the adMSCs. This setup mimics the conditions under which freshly isolated cells from the adipose tissue might be re-transferred to patients in a single surgical procedure, as is being considered for cell therapies [38,39,40]. This model system will require technical modifications before clinical application, such as implementing a flow system to enable continuous cell suspension delivery and to eliminate manual pipetting, thus ensuring compatibility with the time constraints of an operating theater. A single, short-term exposure to three distinct wavelengths (blue: 455 nm, red: 660 nm, and NIR: 810 nm) and their combinations was tested, reflecting the time-sensitive nature of intraoperative cell processing. The wavelengths were selected based on their known specific interactions with cellular chromophores and their documented biological effects [37,41]. In this setting, we examined the effects of light on oxidative stress, viability, proliferation, metabolic activity, migration, and adipogenic differentiation to assess its potential for enhancing the therapeutic efficacy of adMSCs.

## 2. Materials and Methods

### 2.1. AdMSCs Isolation and Cell Culture

Primary adMSCs were isolated from the adipose tissue of nine healthy patients with a mean age of 42.0 ± 12.2 years and a BMI of 34.63 ± 6.60 kg/m^2^ by liposuction. Adipose tissue harvested by liposuction was transported overnight, and the stromal vascular fraction (SVF) was isolated according to the pre-established protocol [42,43]. In brief, the enzymatic digestion utilizing 1.5 U/mL collagenase (NB4 derived from Clostridium histolyticum; Nordmark Biochemicals, Uetersen, Germany) was conducted with gentle agitation for a duration of 30 min at a temperature of 37 °C. Following this, a series of filtration procedures (employing 100 µm and 40 µm cell strainers; Corning, New York, NY, USA), along with washing protocols utilizing phosphate-buffered saline (PBS) supplemented with 10% fetal bovine serum (FBS; both sourced from PAN Biotech, Aidenbach, Germany) and subsequent centrifugation steps ensued. Subsequently, the resultant final cell pellet was resuspended in a culture medium (Dulbecco’s Modified Eagle Medium (DMEM) enriched with high glucose and GlutaMAX) which contained 1% penicillin/streptomycin (both obtained from Gibco by Life Technologies, Darmstadt, Germany) and 10% FBS, then transferred to cell culture flasks for incubation at 37 °C within a humidified environment. Subsequently, the CD34-positive subpopulation of the SVF was purified according to a standardized protocol [42]. 24 h after isolation, the subpopulation of CD34 positive cells was extracted using the Dynal^®^ CD34 precursor cell isolation system (Invitrogen, Karlsruhe, Germany). The non-adherent cells were removed by two washes with PBS and then incubated with magnetic beads conjugated to the CD34 antibody in a cell culture medium. The non-adherent magnetic beads were removed by two further washes with PBS, followed by trypsinization to facilitate cell detachment. A series of magnet exposure and washing steps with PBS/0.1% FBS in a rotary mixer at 4 °C refined the suspension of CD34-positive cells. Afterwards, CD34-positive cells were expanded until passage 2. Cryopreservation of adMSCs was performed according to an established protocol at passage 2 [44]. For all cell culture experiments, adMSCs from passage 2 were thawed according to a standard laboratory protocol and seeded in a 175 cm^2^ flask (Greiner Bio-one, Frickenhausen, Germany) in Dulbecco’s Modified Eagle Medium (DMEM, Gibco, by Life Technologies, Darmstadt, Germany) containing 10% FBS (PAN Biotech, Aidenbach, Germany), 1% penicillin/streptomycin (P/S), and 0.4% GlutaMax™ (Gibco, by Life Technologies, Darmstadt, Germany), from then on referred to as the ‘standard cell culture medium’. All experiments were performed at passage 4.

### 2.2. Light Exposure Setup

A 50 mm diameter integrating sphere (2P4/M, ThorLabs Inc., Newton, NJ, USA) was used as a light exposure device to treat the adMSCs cell suspensions (Figure 1).

The integrating sphere consists of a hollow spherical cavity with an interior coated with a diffusely reflective material, ensuring Lambertian (isotropic) reflection. When light enters the sphere through a peripheral opening, it undergoes multiple diffuse reflections on the inner surface, resulting in a uniform radiance distribution across the sphere’s interior, irrespective of the spatial characteristics of the incident beam. This homogenized light ensures a uniform irradiation of the cell suspension. Three LEDs (blue: M455L4, λ = 445 nm; red: M660L4, λ = 660 nm; infrared: M810L5, λ = 810 nm; ThorLabs Inc., Newton, NJ, USA) were connected to the peripheral openings of the integrating sphere. To ensure comparable light input into the cells, the relationship between the current and radiant power (I-P-curve) of each LED was determined using a power sensor (S130VC, ThorLabs Inc., Newton, NJ, USA) with a USB interface (PM100USB, ThorLabs Inc., Newton, NJ, USA) at the detector port. This procedure ensured a radiant power of 580 mW for all three LEDs. The treatment of the cell suspension (1 × 10^6^ cells in 2 mL) was realized in a cuvette (Sarstedt, Nümbrecht, Germany) in the upper opening of the integrating sphere with wavelength range from 455 nm (blue), 660 nm (red), 810 nm (NIR), and all three wavelengths simultaneously (combined) for 2.5 (2 min. 30 s), 7.5 (7 min. 30 s), and 15 min. This results in radiant energies of 1.45 J, 4.35 J, and 8.7 J (Table 1).

This, in turn, means that with the combined wavelengths, we had a three-fold energy input (3 × 1.45 J and 3 × 4.35 J). At the same time, the input current into the exposure chamber was documented using an ammeter (Voltcraft, M-4650B; Düsseldorf, Germany).

### 2.3. Light Exposure Experiments

Twenty-four hours before the start of the experiment, the standard cell culture medium was replaced with a ‘light irradiation medium’ consisting of DMEM supplemented with 1% P/S and 0.4% GlutaMax™ (Gibco, by Life Technologies, Darmstadt, Germany). Still, the standard cell culture medium did not contain phenol red or FBS. Our initial experiments and previous studies indicated that phenol red and FBS affect light absorption [45,46]. We therefore decided to use a medium without phenol red and FBS during light exposure.

Pre-cultivated adMSCs were exposed to different wavelengths with 1 million cells in 2 mL of exposure medium, with light sources positioned perpendicular to the cell suspension and irradiated as mentioned above (Table 1). Cells not exposed to light in the exposure device served as control cultures. After treatment with the different wavelengths, the temperature (Table 2) was first monitored, and 500 µL was removed to determine cell number and viability (Section 2.4). The remaining cells were seeded in different well plates with an addition of 10% FBS for further analysis.

### 2.4. Measurement of Cell Number and Viability Immediately After Light Exposure

Cell number and viability of adMSCs were quantified immediately after light exposure with different wavelengths, to establish a baseline, ensuring that any delayed changes in proliferation or viability could be attributed to the treatments rather than pre-existing differences. For this purpose, the NucleoCounter^®^ NC-3000™ (Chemometec, Allerod, Denmark) ‘Viability and Cell Count Assay’ was used. Therefore, the previously processed cell suspension (500 µL) was transferred into a 1.5 mL tube (Eppendorf, Hamburg, Germany). The treated cell suspensions were analyzed with volume-calibrated cassettes (Via1-Cassette™) containing two immobilized fluorophores: acridine orange (membrane permeable)—stained all cells—and 4′,6-diamidino-2-phenylindol (membrane impermeable)—stained dead cells—in the Nucleo-Counter^®^ NC-3000™ according to the manufacturer’s instructions for determining cell numbers and viabilities. This fluorescence-based method enables rapid and precise discrimination between live and dead cells in suspension, based on differences in membrane permeability and fluorescent staining. Measurements of the sample were performed twice to increase the accuracy of the calculated cell numbers and viabilities, and the mean was taken.

### 2.5. Quantification of Intracellular Reactive Oxygen Species (ROS)

Intracellular ROS in adMSCs were measured immediately after exposure of the cell suspension to light at different wavelengths. The 6-chloromethyl-2’,7’-dichlorodihydrofluorescein diacetate, acetyl ester (CM-H2DCFDA, Thermo Fischer Scientific, Waltham, MA, USA) was used for this purpose. This dye diffuses passively into the cells, and is converted by intracellular esterases and ROS into a fluorescent molecule that remains in the cell. The fluorescence intensity is directly proportional to intracellular ROS, thus enabling a quantitative determination of ROS production. The DCFDA assay was performed immediately after light exposure due to the known limitation that DCFH is susceptible to direct photo-oxidation by blue light, leading to artificial fluorescence signals [47,48]. Consequently, measured ROS levels after light exposure primarily reflect more stable oxidative species, such as lipid peroxides and residual hydrogen peroxide, rather than the transient reactive oxygen species generated during light exposure.

DCFDA, at a final concentration of 20 µM in pre-warmed phosphate-buffered saline (PBS, PAN Biotech, Aidenbach, Germany) without Ca^2+^ and Mg^2+^, was added to the cells (50,000) immediately after light exposure and incubated at 37 °C for 30 min. After centrifugation at 400× *g* for 5 min, the cells were resuspended with light irradiation medium, transferred into a FACS tube (BD Biosciences, Franklin Lakes, NJ, USA), and measured by the flow cytometer FACS Calibur (BD; argon-ion laser at 488 nm) with CellQuest Pro 4.0.1 (BD) software for data acquisition. The fluorescence intensity of at least 10,000 events was measured and calculated as the FL-2 mean channel (MFI) by FlowJo V. 10.1r1 (FlowJo, LLC; BD Becton Dickinson and Company, Franklin Lakes, NJ, USA). Hydrogen peroxide (5% H_2_O_2_; Sigma-Aldrich, St. Louis, MO, USA) served as a positive control [49].

### 2.6. Determination of Metabolic Activity by Tetrazolium Salt Assay

The metabolic activity of adMSCs exposed to different wavelengths was quantified using the Cell Titer 96 Aqueous One Solution Cell Proliferation Assay (MTS, Promega, Walldorf, Germany) 24, 48 h, and 17 days after light exposure. The term ‘metabolic activity’ refers specifically to the measurement of mitochondrial enzyme-mediated substrate reduction. For this purpose, the cells were incubated for 90 min with MTS (3- (4,5-dimethylthiazol-2-yl)-5- (3-carboxymethoxyphenyl)-2- (4-sulfophenyl)-2H-tetrazolium) solution as previously published [50]. The MTS reagent is reduced by NAD (P)H-dependent cellular oxidoreductase enzymes inside the cells. This colored formazan product is directly proportional to the metabolic activity of the cells. The absorbance was measured in a microplate reader (TECAN, Männedorf, Austria) at 490 nm and a reference wavelength of 620 nm.

### 2.7. Determination of Cell Number and Population Doubling Time (PDT)

After determining the metabolic activity, the cell number was quantified using Hoechst H33342 (Applichem, Darmstadt, Germany) 24 and 48 h after light exposure. For this purpose, adMSCs were washed with PBS without Ca^2+^ and Mg^2+^ and then fixed with 4% paraformaldehyde (PFA; Sigma-Aldrich, St. Louis, MO, USA). After washing three times, the cells were incubated with Hoechst 33342 (5 µg/mL) for 15 min in the dark. The fluorescence microscopic images were acquired with the Hermes WiScan system (IDEA Bio-Medical, Rehovot, Israel) and analyzed with the Athena software V1.0.10.116 (IDEA Bio-Medical). Additionally, light-exposed adMSCs were seeded directly after treatment in a 48-well plate with an initial cell density of 38,000 cells/well to determine the population doubling time. Cell numbers were continuously detected from day 1 to day 6 using Hoechst H33342 staining. Fluorescence microscopic images were acquired with the Hermes WiScan system, processed with Fiji ImageJ version 2 (National Institutes of Health, Bethesda, MD, USA), and calculated with the following formula [51]:$$\mathrm{PDT}\;=\;\frac{cultivation\;time\;\times \mathrm{lg}2}{(\mathrm{lg}\mathrm{final}\;cell\;number-\mathrm{lg}initail\;cell\;number}.$$

### 2.8. Determination of Mitochondrial Respiration

To determine the oxygen consumption rate (OCR) and the extracellular acidification rate (ECAR) after light exposure with different wavelengths in the adMSCs, 15,000 cells/well were first cultured in special 8-well plates for 24 and 48 h. According to the manufacturer’s and a previously published protocol, measurements were performed with the Seahorse XF HS Mini Analyzer (Agilent Technologies, Santa Clara, CA, USA) [42].

### 2.9. Measurement of IL-6 and IL-8 by ELISA

To understand the effects of PBM on immune regulation, the total protein concentration and the concentrations of the pro-inflammatory cytokines interleukin (IL)-6 and IL-8 were determined. For this purpose, the supernatants were collected 24 h after exposure to adMSCs with different wavelengths in low-binding tubes (Sarstedt, Nümbrecht, Germany) and centrifuged at 12,000× *g* for 7 min to remove the cellular debris. The supernatant was then stored at −80 °C until further analysis. According to the manufacturer’s instructions, the total protein concentration was determined from 5 µL of collected and stored supernatant using a DC protein assay kit (BIO-RAD Laboratories GmbH, Munich, Germany), and pro-inflammatory cytokines IL-6 and IL-8 were quantified using ELISA (R&D System, Inc., Minneapolis, MN, USA). Absorbance was measured using a microplate reader (TECAN) with 650 nm for protein concentration and 450 nm with a reference wavelength of 540 nm for IL-6 and IL-8. The concentrations of IL-6 and IL-8 were measured in pg/mL and normalized to the respective cell number.

### 2.10. Analysis of Cell Migration

Cell migration was measured using the scratch assay by seeding the light-exposed adMSCs at a cell density of 11,000 cells/well in a 96-well plate. At 24 h after seeding, standard cell culture media were replaced with a cell culture medium without phenol red containing a red cell tracking dye (Abcam, Cambridge, UK) at a dilution of 1:60 according to the published protocol [52]. After a further 24 h of cultivation, a wound was created using a wound maker (Incucyte; Sartorius, Göttingen, Germany) according to the manufacturer’s instructions, and a first image was acquired at time 0 (T0) using the WiScan system. A second series of images was retaken with the WiScan system 15 h after wound creation. Wound closure was quantified using Fiji ImageJ software, and the percentage of wound closure was calculated using the following formula:$$\frac{initial\;wound\;area-\mathrm{final}\;wound\;area}{inital\;wound\;area}\times 100\%$$

### 2.11. Determination of Adipogenic Differentiation Capacity of adMSCs upon Light Exposure to Different Wavelengths

Light-exposed adMSCs cells were initially cultivated on a 6-well plate for three days to assess adipogenic differentiation. Subsequently, the cells were reseeded into a 96-well plate to ensure a uniform starting cell density for the differentiation experiments. The 96-well plate was then cultured for an additional three days to allow the cells to reach confluence before initiating adipogenic stimulation (six days after light exposure). Adipogenic differentiation was induced, as previously published [29]. Control cultures consisted of adMSCs maintained in a standard culture medium without adipogenic supplements to facilitate a comparative analysis of adipogenic differentiation.

Lipid accumulation as proof of adipogenic differentiation was assessed after 14 days of adipogenic stimulation using a lipid-specific fluorophore (Bodipy: boron-dipyrromethene with 1 µg/mL in 150 mM NaCl, Life Technologies) according to a published protocol [29]. Additionally, cell nuclei were stained with Hoechst 33342. Fluorescence imaging was performed using the Hermes WiScan system, and subsequent analysis was conducted with Athena software. The fluorescence intensity of Bodipy staining was normalized to the cell number determined via Hoechst H33342 staining.

### 2.12. Statistical Analyses

All experiments were performed with three to five independent donors, each in triplicate. Data were visualized and statistically analyzed with Microsoft Excel 2010 and GraphPad Prism, Version 7 (GraphPad Software Inc., San Diego, CA, USA). Data in Figures 2 and 4b were presented as bars (mean) with standard deviations and in Figures 3–8 and Figure A1 as boxplots with means (+). The horizontal line within the box plot indicates the median, with whiskers showing the minimum and maximum data points. Statistical significance was calculated using the Kruskal–Wallis test in the absence of normal distribution (Shapiro–Wilk test) or Ordinary one-way ANOVA in the presence of normal distribution (Shapiro–Wilk test). The significance level was set at *p* ≤ 0.05 (*).

### 2.13. Ethical Statement

All tissue samples analyzed were collected with the patient’s consent. The local ethics committee approved using the tissue samples under registration no. A2020-0054 and A2025-0012.

## 3. Results

### 3.1. Detection of Intracellular Reactive Oxygen Species (ROS) in adMSCs upon Light Exposure

To investigate the generation of ROS elicited by light exposure, we quantified intracellular ROS levels in adMSCs immediately following the exposure of the cell suspensions to light at diverse wavelengths and radiant energy. This quantification was achieved using an ROS-sensitive dye combined with flow cytometric analysis. Light exposure across all examined wavelengths showed a notable increase in intracellular ROS levels. Importantly, exposure to blue light treatments elicited statistically significant increases in ROS levels compared to untreated control cultures (Figure 2).

### 3.2. Impact of Light Exposure at Different Wavelengths on the Number, Viability, and Metabolic Activity of adMSCs

Cell number and viability of adMSCs were quantified immediately after light exposure. No significant changes in either cell number (Figure 3) or viability (Table A1) were observed across the various light exposure parameters examined. The viability of the cells remained uniformly elevated, ranging from 92.8% to 95.7%, irrespective of the wavelength utilized.

After seeding, the blue light-exposed adMSCs and their subsequent 24 h cultivation led to a pronounced, wavelength-dependent reduction in cell number (Figure 4a). Simultaneous exposure to multiple wavelengths (combined) also reduced cell number, though the effect was less pronounced compared to that induced by isolated blue light. In contrast, red or NIR light exposure did not affect cell number.

These observations were further confirmed by quantifying the population doubling time over 5 days. Exposure to blue light significantly increased the population doubling time, indicating decreased proliferative activity (Figure 4b).

Overall, the observed reduction in cell number after 24 h of exposure and the prolonged population doubling time indicate that blue light induces a cytotoxic effect and impairs the proliferative activity of adMSCs. In contrast, neither red nor NIR light showed any discernible impact on viability or proliferation.

Similar to the changes in cell numbers after exposure to different wavelengths, the metabolic activity, as assessed by a tetrazolium salt assay, was also significantly and dose-dependently reduced by treatment with blue light (Figure 5a). Conversely, the other wavelengths examined did not affect the metabolic activity of adMSCs.

Notably, exposure to blue light resulted in a non-statistically significant increase in the basal respiration of adMSCs (Figure 5b). Although the results did not reach statistical significance, this observation is interesting as it contradicts the results of the tetrazolium salt test. No significant effects on basal respiration were observed following exposure to other wavelengths of light.

### 3.3. Effect of Light Exposure on Pro-Inflammatory Cytokine Release of adMSCs

The influence of various light exposures on the inflammatory response was assessed by quantifying the pro-inflammatory cytokines interleukin-6 (IL-6) and IL-8 in cell culture supernatants. Exposure to blue light resulted in a radiant energy-dependent decrease in the release of both IL-6 (Figure 6) and IL-8 (Figure A1) from adMSCs, compared to control (normalized to cell number). In contrast, no significant changes were observed after red or NIR light exposure.

### 3.4. Analysis of the Migration Activity in Light-Exposed adMSCs

The migration activity of adMSCs was analyzed using the scratch assay, which assesses the cells’ ability to migrate and cover an in vitro wound. The exposure of adMSCs to blue light did not induce significant effects on migratory activity compared to the untreated control cultures (Figure 7). In contrast, treatment with red and NIR light significantly enhanced migration in the wound healing model. Notably, NIR light exhibited a significant, radiation energy-dependent effect on cell migration.

### 3.5. Persistent Effects of a Single Light Exposure on Cell Number and Adipogenic Differentiation

Consistent with the results observed shortly after light exposure, the blue light-induced reduction in cell number continued for more than two weeks in a dose-dependent manner. This reduction indicates a persistent impairment in the proliferation capacity of adMSCs treated with blue light (Figure 8a). In contrast, other wavelengths did not induce significant effects, except in the case of combined light treatment, which is likely attributable to the influence of blue light. At this later stage (i.e., approximately 17 days after light exposure), lipid accumulation as a proof of adipogenic differentiation capacity of light-treated adMSCs was examined. Exposure to blue light led to a significantly reduced adipogenic differentiation compared to untreated control cultures (Figure 8b,c). In contrast, no significant differences in adipogenic differentiation were observed in adMSCs exposed to red or NIR light.

## 4. Discussion

PBM has proven beneficial in various areas of regenerative medicine, such as accelerating tissue repair after surgery, improving the healing of chronic wounds, and reducing inflammatory processes. However, the clinical effects of PBM appear to depend on factors such as wavelength, dosage, energy output, and type of light emission (pulsed or continuous). The underlying mechanisms are not yet fully understood, and individual factors such as skin characteristics further complicate our understanding [14]. To fill these knowledge gaps, we investigated the effects of short-term exposure to different wavelengths—blue (455 nm), red (660 nm), and NIR (810 nm)—on the immediate and longer-term behavior of adMSCs. In contrast to conventional in vitro light exposure studies, where irradiation is performed on adherent cell layers, we utilized an approach in which cells were irradiated in suspension within a tube in an integrating sphere. This method is expected to provide more homogeneous light exposure compared to fixed light sources for adherent cultures, which are prone to shading effects and light gradients [53,54,55,56]. Notably, in our protocol, cells were seeded immediately after irradiation. Therefore, the subsequent cellular behavior reflects both the immediate effects of light exposure in a simplified environment and the restoration of normal cell–matrix interactions after seeding. This approach can regarded as a model system but may the outcome be particularly relevant for applications such as intraoperative cell therapy.

### 4.1. Light Exposure Induces Reactive Oxygen Species (ROS) Accumulation

ROS are highly reactive oxygen-derived molecules generated as by-products of normal cellular metabolism, primarily within the mitochondria [57]. At physiological levels, ROS function as essential signaling molecules, regulating cell proliferation, differentiation, migration, immune responses, and programmed cell death [58]. However, excessive accumulation of ROS disrupts this balance, resulting in oxidative stress that impairs cellular functions and causes molecular damage [59].

Endogenous light-sensitive molecules such as cytochrome C oxidase and flavins are critical mediators of light-induced cellular responses [13,60]. These chromophores influence the formation of ROS when exposed to different wavelengths [60]. In our study, all tested wavelengths elevated ROS in adMSCs due to the different light exposure, with blue light causing the strongest effect. In the blue light range, flavins and flavoproteins in mitochondrial complexes I and II are selectively activated, boosting electron transport chain activity [5]. This surge in ROS production and the associated oxidative stress can impair mitochondrial integrity, activate apoptosis via caspase pathways, and stimulate the tumor suppressor protein p53 [60,61]. These processes underscore the dual nature of ROS—vital for homeostasis at baseline levels, yet destructive when dysregulated—and highlight the unique potency of blue light in disrupting cellular balance. Such light-induced oxidative stress might impair the function of adMSCs, as oxidative stress is known to reduce both their viability and therapeutic efficacy [52,62].

Consistent with our findings, Wang et al. demonstrated that exposing adMSCs to different light wavelengths (415 nm, 540 nm, 660 nm, and 810 nm) increases ROS generation at all wavelengths with blue light causing the highest ROS levels and cytotoxicity [37]. They also reported that NIR light enhances mitochondrial efficiency and upregulates antioxidant enzymes, potentially preventing redox imbalance [63]. Cytochrome c oxidase, a key mitochondrial chromophore, absorbs red and NIR light, which can improve mitochondrial respiration and ATP production and induce moderate ROS signaling to enhance cellular resilience [64,65]. However, there is the suggestion that cytochrome c oxidase may not be the primary NIR receptor [66], and prolonged NIR exposure can lead to higher ROS levels, highlighting the complexity of wavelength and exposure duration [67]. In summary, blue light and the associated increase in oxidative stress could impair the function of adMSCs through mitochondrial dysfunction, while red and NIR light have comparatively little effect on ROS levels. Understanding these ROS dynamics is crucial for minimizing light-induced damage. Differences in experimental parameters probably explain the discrepancies in the ROS levels and the oxidative stress responses between the studies, complicating direct comparison.

### 4.2. Effects of Blue Light on adMSCs Proliferation and Metabolism

Immediately after light exposure, none of the examined wavelengths affected cell viability or number. However, after 24 h of cultivation, blue light (455 nm) caused a dose-dependent cytotoxic effect as reflected by a reduced cell number. Furthermore, a prolonged PDT, indicating a slowdown of the cell cycle, could be observed within the following 5 days. Whether this slowing affects the entire cell population remains unclear. Combinations of wavelengths that included blue light also decreased cell number after 24 h. In contrast, exposure to red or NIR light did not affect cell number or PDT. These findings align with previous reports showing that blue light inhibits cell proliferation and subsequently induces cell death, often via apoptosis, likely due to increased ROS production [37].This can lead to oxidative damage, inhibition of cell proliferation, and activation of apoptotic pathways, such as inactivation of protein kinase C orcaspase-3 activation, as well as DNA damage via the H2A.X-Nox1/Rac1 signaling pathway, as suggested by several studies [68,69,70,71,72]. The observed 24 h delay in cell number reduction matches the time course of apoptotic processes (e.g., caspase activation, DNA fragmentation, and mitochondrial dysfunction) [73]. The prolonged PDT following blue light exposure suggests a ROS-induced cell cycle arrest, which has also been reported in other cell types, such as bone marrow-derived MSCs and porcine retinal cultures [72,74].

Interestingly, while Wang et al. reported enhanced cell proliferation under red and NIR light [37], we did not observe this effect. This discrepancy may be explained by significant differences in experimental design such as adherent 2D cultures versus adMSCs in suspension, variations in cell culture format and illumination parameters, which are known to influence cellular responses to light [75].

Furthermore, we demonstrated that the various wavelengths differentially affected adMSCs’ energy metabolism. Red and NIR light had no effect on the energy metabolism of adMSCs, while exposure to blue light resulted in decreased overall metabolic activity (as measured by tetrazolium salt assay) but a slight, non-significant increase in oxygen consumption rates (OCRs) during mitochondrial stress testing.

Tetrazolium salt assays, such as MTS, measure cellular metabolic activity by quantifying the reduction of MTS to formazan, a process driven by NAD (P)H-dependent enzymes present in viable cells [76]. The amount of formazan produced correlates with the metabolic activity of the cells: a decrease in metabolic activity results in lower formazan production and a reduced absorbance reading. Thus, if adMSCs exposed to blue light show less formazan, this indicates a decrease in their overall metabolic activity. In contrast, OCR specifically reflects mitochondrial respiration and ATP production [77]. An increase in OCR typically signals enhanced oxidative phosphorylation, as cells consume more oxygen to generate ATP, increased OCRs typically indicate enhanced oxidative phosphorylation activity, a process where cells utilize oxygen to produce ATP [78]. However, it is possible for overall metabolic activity to decrease while OCR increases, particularly in cases of mitochondrial uncoupling—where oxygen consumption is elevated without a proportional rise in ATP production [79,80]. Additionally, cells may temporarily increase their respiration rate as a stress induced compensatory response to maintain energy balance or adapt to damage [77]. This response is often short-lived and does not necessarily reflect an overall or sustained metabolic activity increase. Another consideration is the possibility of metabolic pathway shifts (e.g., from glycolysis to fatty acid oxidation), leading to higher OCRs despite reduced viability [81,82]. These findings highlight the importance of interpreting OCR and metabolic assays in context of cellular physiology, especially when evaluating responses to experimental treatments such as light exposure. In summary, blue light exposure specifically impacts adMSCs’ proliferation and energy metabolism, whereas red and NIR light do not, even at the highest tested doses.

### 4.3. Differential Effects of Light Wavelengths on Pro-Inflammatory Cytokine Release of adMSCs

PBM is increasingly recognized for its capacity to modulate inflammatory responses, supporting its therapeutic applications in regenerative medicine [83]. The PBM’s anti-inflammatory effects depend on light wavelength and exposure parameters, acting through regulation of oxidative stress, transcription factors such as NF-κB, and the downregulation of pro-inflammatory cytokines [14].

Our findings and recent studies highlight the distinct immunomodulatory effects of different PBM wavelengths. Blue light has shown strong potential to reduce tissue inflammation and to promote wound healing [84,85]. In our experiments, exposure of adMSCs to blue light (455 nm) resulted in a significant decrease in the release of the pro-inflammatory cytokines IL-6 and IL-8. These observed anti-inflammatory effects of blue light, also described in previous studies, may be associated with suppression of NF-κB or MAPK activity, potentially through reactive species-dependent signaling mechanisms [86]. Consequently, exposure to blue light could help to reduce pro-inflammatory responses, which could benefit autoimmune or delayed healing processes. However, reduced cytokine levels might also result from nonspecific effects, such as impaired cellular metabolism or apoptosis, emphasizing the need for further studies.

In contrast, the irradiation of adMSCs with red and NIR light did not significantly alter the release of IL-6 or IL-8. This finding diverges from some studies, where red and NIR light have demonstrated anti-inflammatory effects in other cellular models. Studies using 808 nm lasers light have reported reductions in IL-6 and TNF-α levels in human bone marrow-derived MSCs. Similar effects have been observed in LPS-treated animal models, where these wavelengths decrease pro-inflammatory cytokines by modulating NF-κB and inflammasome signaling pathways [87,88]. However, these anti-inflammatory effects of red and NIR light are more evident in cells that are already immune-primed or inflamed, such as LPS-stimulated MSCs, rather than naïve/unstimulated adMSCs [83,89]. These discrepancies may be explained by differences in experimental conditions, such as the specific cell type or animal model used, the irradiation protocol, and the timing of cytokine measurement.

Mechanistically, red and NIR light are thought to act via mitochondrial signaling (e.g., cytochrome c oxidase activation, ATP elevation), suppressing NF-κB activity, while blue light acts primarily through ROS-dependent pathways [83,89].

In summary, PBM’s immunoregulatory effects are complex and context-dependent requiring careful consideration of wavelength, dosage, and biological context to achieve optimal immunomodulatory clinical outcomes.

### 4.4. Migration of adMSCs Is Enhanced by Exposure to Red and NIR Light

The migration of cells is a fundamental aspect of tissue regeneration, allowing MSCs to move to sites of injury or damage [90]. This process is regulated by factors such as ROS, inflammatory signals, extracellular matrix composition, cell–cell interactions, and mitochondrial function [91,92]. In our study, exposure to red and NIR light significantly enhanced the migration of adMSCs, as evidenced by accelerated wound closure in vitro. This is consistent with previous reports demonstrating that 660 nm and 810 nm wavelengths increase ATP production, mitochondrial membrane potential, and upregulate migration-associated markers such as chemokine receptor type 4 (CXCR4) [67].

Mechanistically, red and NIR light stimulate cytochrome c oxidase within the mitochondrial respiratory chain, increasing ATP synthesis [5,35,93], which supports actin cytoskeletal reorganization that is crucial for migration [94]. These wavelengths also help modulate ROS, reduce oxidative stress, and preserve cellular function. Notably, our results indicate that 810 nm NIR light is more effective than 660 nm red light in promoting adMSCs migration, likely due to its greater activation of mitochondrial processes [95].

Clinically, preconditioning adMSCs with red and NIR light may enhance their migratory abilities and therapeutic potential in conditions requiring efficient stem cell homing.

### 4.5. Persistent Impact on Proliferation and Lipid Accumulation

Our experiments demonstrated that a single short-term exposure of adMSCs to blue light or a combined wavelength containing blue light results in a sustained impairment of both proliferation and metabolic activity, detectable even 17 days post-irradiation. While previous studies mostly reported these inhibitory effects of blue light on cell proliferation and metabolism within a few days (up to three), recent research and our findings suggest that these detrimental effects can persist much longer, particularly with repeated exposure. Palumbo (2020) reported that a daily irradiation with 453 nm light reduced metabolic activity in 3T3-L1 preadipocytes by about 10% after 14 days [96]. Despite differences in cell type and irradiation protocols, these findings align with our observations, suggesting that blue light can exert harmful effects under various conditions. The impairment upon blue light exposure is primarily linked to mitochondrial dysfunction, as blue light reduces ATP production and increases ROS formation, causing oxidative stress and energy depletion [10,37,97]. Notably, our data and the literature indicate that red and NIR light tend to promote cell growth and survival or at least do not impair these parameters [37,83,98].

The multipotency of adMSCs, which enables differentiation into adipocytes, chondrocytes, and osteoblasts, is central to their regenerative potential [27,28,29,99] and can be influenced by various external factors such as light/irradiation. Although blue light is known to reduce osteogenic differentiation capacity, little is known about its effect on adipogenesis, whereas red and NIR light generally do not have adverse effects [100,101]. Our results extend these findings to adipogenesis: Blue light at 455 nm caused a dose-dependent reduction in lipid accumulation during differentiation. This is also supported by the work of Palumbo, showing dose-dependent inhibition of adipogenic differentiation in 3T3-L1 preadipocytes after repeated blue light exposure during differentiation [96]. This consistency implies that blue light’s inhibitory effects on adipogenic differentiation may be applicable across mesenchymal cell types. This inhibition is likely due to blue light-induced mitochondrial dysfunction and elevated oxidative stress, which impair the energy-intensive process of adipogenesis [97]. In contrast, red and NIR wavelengths appeared neutral regarding adipogenic differentiation in our experiments, likely due to their lack of detrimental effects on mitochondrial function and ATP production [63,83].

## 5. Conclusions

Integrating PBM with (ad)MSC-based therapies holds considerable potential for regenerative medicine. In this study, adMSCs were illuminated in an integrative sphere to ensure uniform light exposure, a method designed for clinical translation. Exposure to 455 nm blue light negatively affected adMSCs, reducing their metabolic activity and differentiation potential while increasing oxidative stress, which could compromise their therapeutic efficacy. Conversely, PBM with red or NIR light enhanced adMSC migration, supporting its use in wound healing and tissue regeneration. The minimally invasive and low-cost nature of PBM further highlights its clinical promise. Through this study we have established a foundation to develop point-of-care protocols wherein adMSCs are isolated (e.g., via liposuction), exposed briefly to optimized PBM wavelengths to enhance their regenerative profile, and then reintroduced locally into the wound environment. However, challenges remain due to a lack of standardized protocols for wavelength, dosage, and exposure duration.

Future research should focus on optimizing and standardizing light parameters for suspension-based irradiation, and validating these parameters in preclinical models. By combining innovative light delivery with adMSCs biology, this work advances the development of practical and scalable regenerative therapies.

## Figures and Tables

**Figure 1 cells-14-01143-f001:**
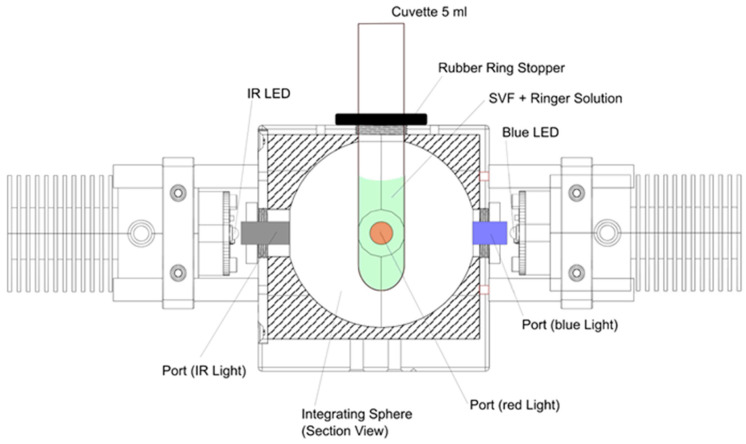
Schematic representation of the light exposure device and the treatment of the cell suspension. A cuvette containing the cell suspension is positioned vertically at the top center of the integrating sphere. The LEDs used for illumination are mounted on various sides of the sphere, directing their light into the sphere through designated ports. The light is diffusely reflected within the sphere, resulting in a homogenized irradiation of the cell suspension.

**Figure 2 cells-14-01143-f002:**
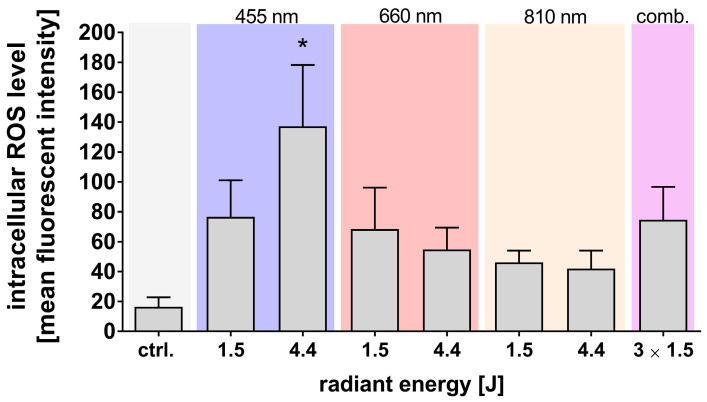
Quantification of intracellular ROS after light exposure. Intracellular ROS levels in adMSCs were measured immediately after exposure to different wavelengths (455 nm, 660 nm, 810 nm, and combined/comb.) using the fluorescent probe CM-H_2_DCFDA. Data are shown as means ± SD, with statistical significance (* *p* ≤ 0.05) determined by ordinary one-way ANOVA and Dunnett’s post hoc test; n ≥ 3. Comparisons were made against untreated controls (ctrl.).

**Figure 3 cells-14-01143-f003:**
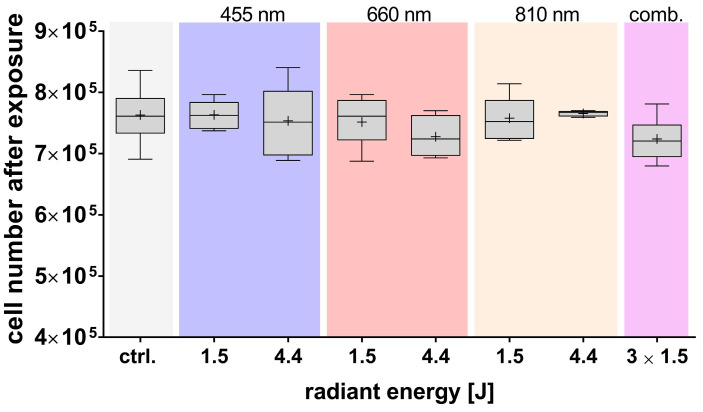
Immediate effects of light exposure on adMSCs cell number. The number of adMSCs was analyzed immediately after light exposure at wavelengths of 455 nm, 660 nm, and 810 nm, and their combination (comb.). Data are presented as boxplots showing mean values (+), medians, interquartile ranges, and minimum/maximum values. The normal distribution was confirmed using the Shapiro–Wilk test, and statistical significance was determined by ordinary one-way ANOVA with Dunnett’s post hoc test, which was not significant; n ≥ 3.

**Figure 4 cells-14-01143-f004:**
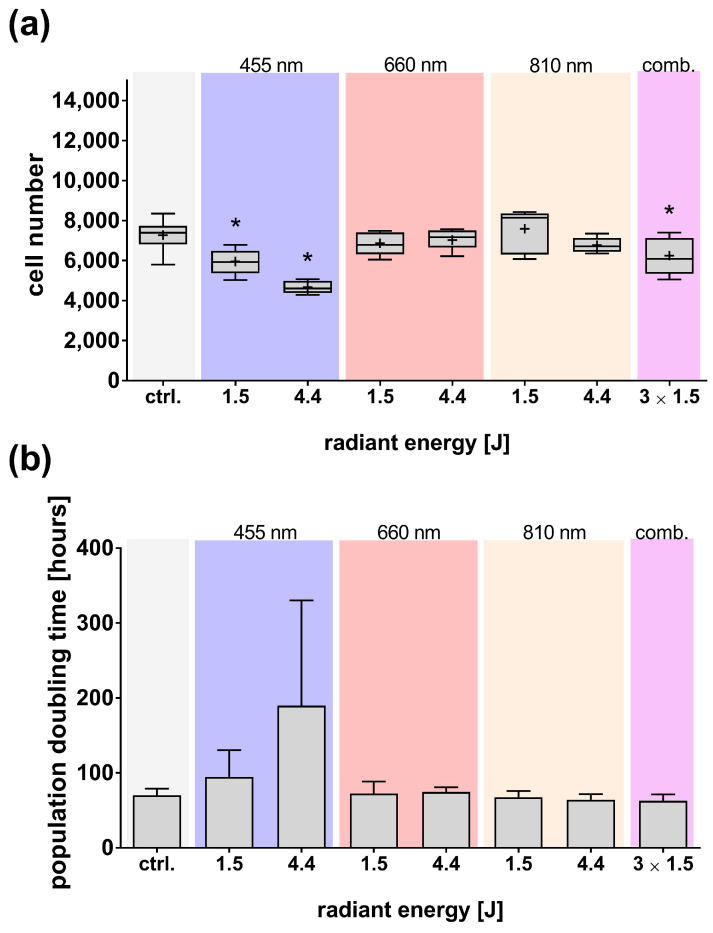
Effects of light exposure (455, 660, 810 nm, and a combination/comb.) on adMSCs proliferation and population doubling time. (**a**) Cell numbers were analyzed after 24 h (boxplots showing mean values (+), medians, interquartile ranges, and minimum/maximum values), and (**b**) population doubling time was assessed over 5 days following light exposure (both by quantifying nuclei via image analysis; mean ± SD). Statistical significance was determined based on the normal distribution (Shapiro–Wilk test) using either ordinary one-way ANOVA with Dunnett’s post hoc test or the Kruskal–Wallis test with Dunn’s post hoc test. The significance level was set at * *p* ≤ 0.05 compared to the untreated control (ctrl.); n ≥ 3.

**Figure 5 cells-14-01143-f005:**
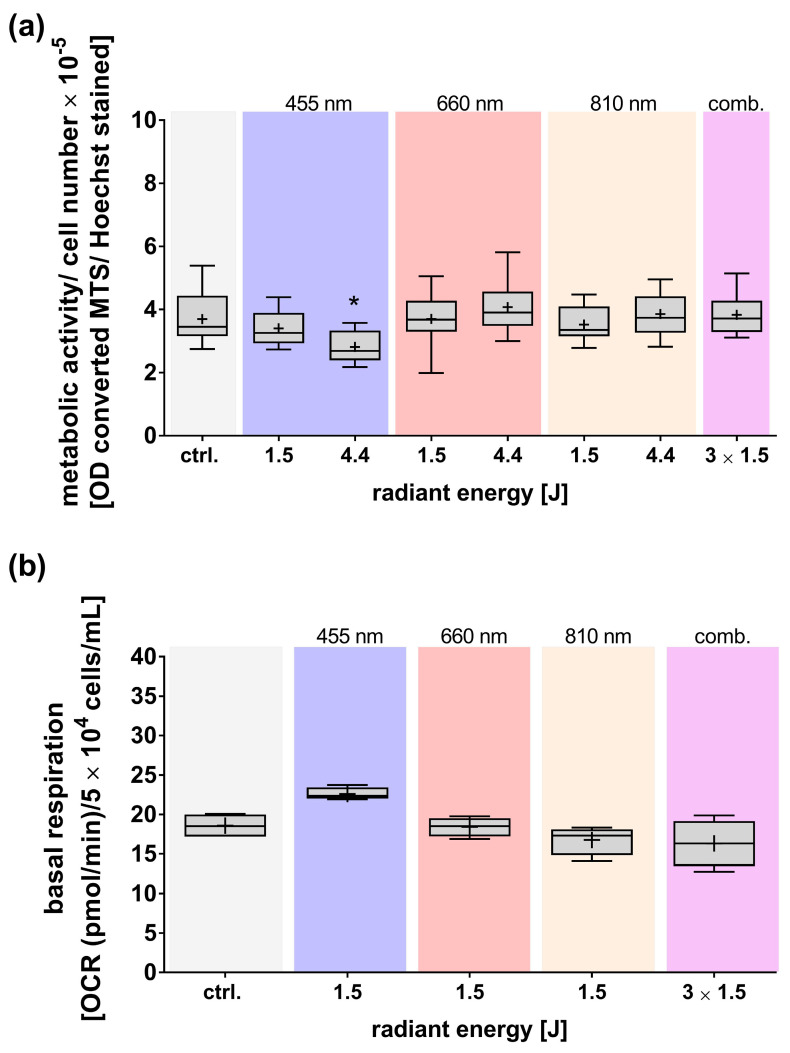
Effects of light exposure on the metabolic activity of adMSCs. Analyses were conducted 24 h after light exposure at different wavelengths (455, 660, 810 nm, and a combination/comb. of all). (**a**) Relative metabolic activity was quantified by normalizing the results to the respective cell numbers (based on the conversion of a tetrazolium compound). (**b**) Analysis of the basal oxygen consumption rate. Data are presented as boxplots showing mean values (+), medians, interquartile ranges, and minimum/maximum values. Statistical significance was determined based on data distribution (Shapiro–Wilk test), using ordinary One-way ANOVA with Dunnett’s multiple comparison post hoc test or the Kruskal–Wallis test with uncorrected Dunn’s multiple comparison post hoc test. Significance (* *p* ≤ 0.05) was compared to the untreated control (ctrl.); n ≥ 3.

**Figure 6 cells-14-01143-f006:**
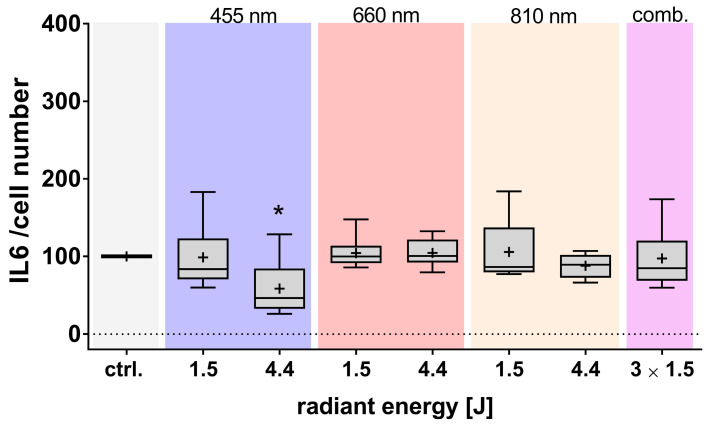
Effect of light exposure on interleukin (IL)-6 release from adMSCs. The IL-6 concentration was analyzed by ELISA 24 h after light exposure with different wavelengths (455, 660, 810 nm, and a combination of all three wavelengths/comb.). The data were normalized to cell number and are presented as boxplots showing mean values (+), medians, interquartile ranges, and minimum/maximum values. The normal distribution was verified using the Shapiro–Wilk test. Statistical significance (* *p* ≤ 0.05) was determined using the Kruskal–Wallis test followed by uncorrected Dunn multiple comparison post hoc test; comparison to the untreated control (ctrl.); n ≥ 3.

**Figure 7 cells-14-01143-f007:**
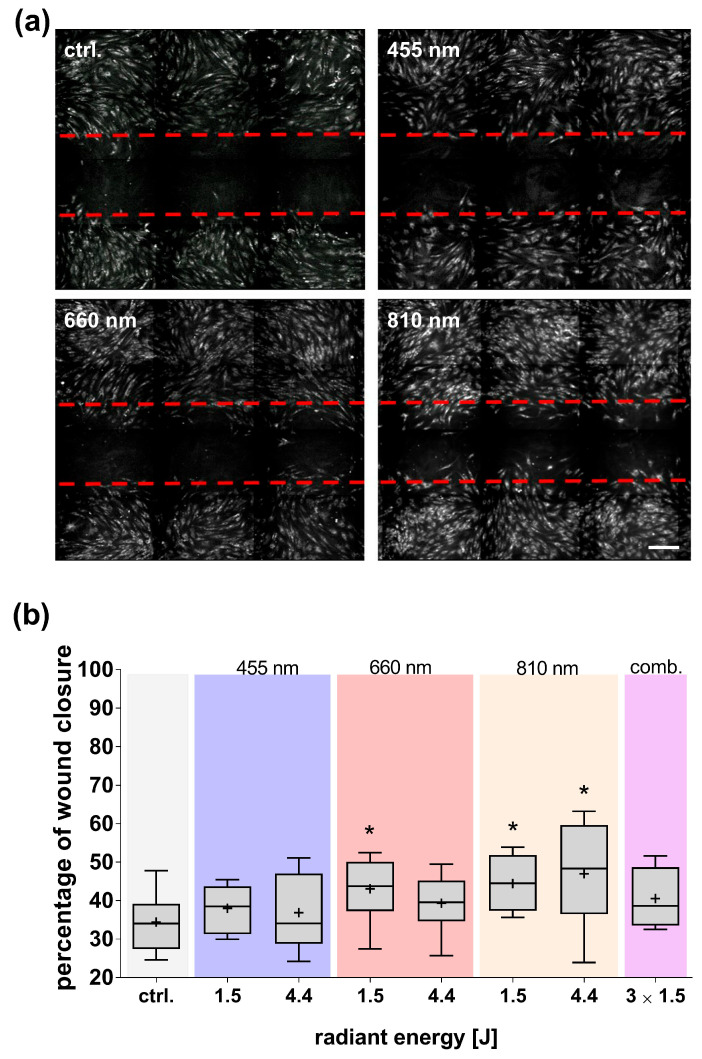
Impact of light exposure on the migratory activity of adMSCs. (**a**) Representative images of adMSCs wound closure after light exposure [4.4 J] (scale bar: 50 µm). (**b**) Image analysis quantified wound closure after light exposure at different wavelengths (455, 660, 810 nm and light of all three wavelengths combined/comb.). The quantification of cell overgrowth was performed 15 h after scratching. The data set is presented as boxplots with mean values (+), medians, interquartile ranges, and minimum/maximum values. The normal distribution was tested using the Shapiro–Wilk test. Statistical significance (* *p* ≤ 0.05) was determined using the Kruskal–Wallis test and an uncorrected Dunn post hoc test for multiple comparisons; comparison to the untreated control (ctrl.); n ≥ 3.

**Figure 8 cells-14-01143-f008:**
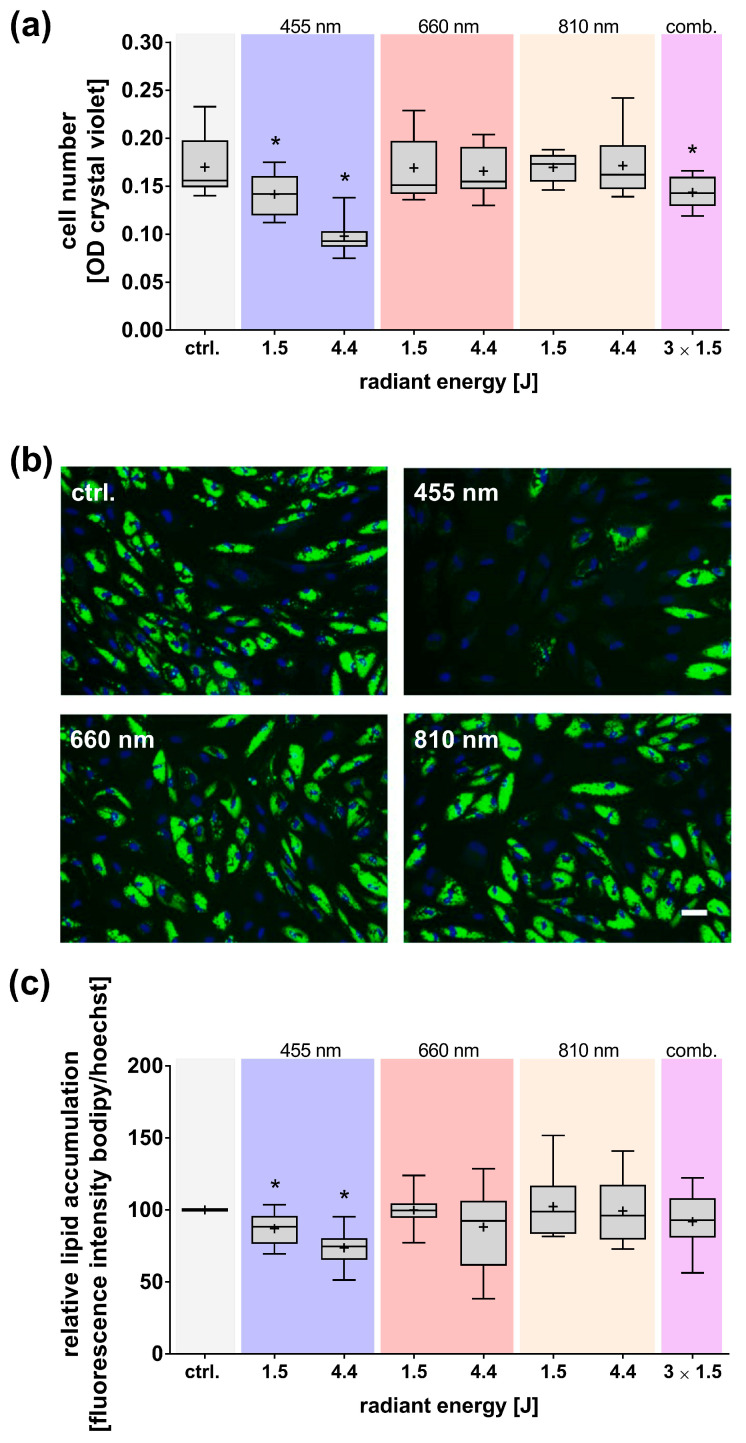
Analysis of the effects of light exposure on cell number and adipogenic differentiation of adMSCs 17 days after light exposure at different wavelengths (455, 660, 810 nm, and a combination of all three wavelengths). (**a**) Quantification of adMSCs cell number, (**b**) microscopic images showing lipid accumulation (green) and nuclei (blue, light exposure at 4.4 J, scale bar: 50 µm), and (**c**) quantification of relative lipid accumulation in adMSCs, normalized to cell number. Data are presented as boxplots showing the mean (+), median, interquartile range, and minimum/maximum values. The normal distribution was assessed using the Shapiro–Wilk test. Statistical significance (* *p* ≤ 0.05) was determined using either ordinary one-way ANOVA followed by Dunnett’s multiple comparison post hoc test or the Kruskal–Wallis test followed by Dunn’s uncorrected multiple comparison post hoc test compared to control (ctrl.); n ≥ 3.

**Table 1 cells-14-01143-t001:** Calculated radiant energy for 455, 660, 810 nm, and all wavelengths combined according to the treatment time.

Exposure Time [Min]	Radiant Energy [J]
2.5	1.45
7.5	4.35
15	8.70

**Table 2 cells-14-01143-t002:** Temperature measurements after exposure to light.

Wavelength [nm]	Radiant Energy [J]	Temperature [°C] After Treatment
without	Control (ctrl.)	26.6
455 Blue	1.45	26.8
	4.35	31.8
660 Red	1.45	23.7
	4.35	26.0
810 NIR	1.45	24.0
	4.35	26.2
455–810 Combined	3 × 1.45	27.1
	3 × 4.35	35.8

## Data Availability

The data presented in this study are available in the article and from the corresponding author.

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
