# Peer review of "Light Exposure as a Tool to Enhance the Regenerative Potential of Adipose-Derived Mesenchymal Stem/Stromal Cells"

_cells, 2025, doi:10.3390/cells14151143_

Round 1
Reviewer 1 Report
Comments and Suggestions for Authors
General considerations
The authors present a manuscript describing the use of light exposure as a tool to enhance the regenerative potential of adipose-derived mesenchymal stem cells (adMSCs). They examined the impact of a single, short-term exposure to three wavelengths and their combination at different dosages on critical cellular parameters such as oxidative stress, viability, proliferation, and adipogenic differentiation potential, among others. They show that light exposure triggers various reactions in the adMSCs, mainly affecting intracellular ROS levels, metabolic activity, proliferation, and migration dynamics.
The overall approach is well-structured and interesting. The experimental results presented in the manuscript support the initial hypothesis that photobiomodulation-activated adMSCs may accelerate wound healing by improving cell migration following red or NIR irradiation.
Apart from a few oversights, the manuscript is logical and well-written. I strongly recommend publication after minor revisions.
Minor revisions:
Page 3, line 98: The sentence is grammatically incorrect. Please revise.
Page 3, paragraph 2.1: Use “at passage x” instead of “in passage x”. The error occurs twice. Please correct it.

Reviewer 2 Report
Comments and Suggestions for Authors
I had the opportunity to review the scientific work related to ADMS regenerative potential. Below are my constructive comments for authors
- the introduction section should describe/detail on why the chosen wavelengths (455 nm, 660 nm, 810 nm) were specifically relevant for adMSCs
- why phenol-red and serum-free medium were selected? A good idea would be to cite previous work/ recommendations
- address cell suspension vs. adherent cells vs. short exposure time
- why increased ROS after blue light did not affect intracellular calcium levels?
- detail on the unexpected metabolic assay (tetrazolium) vs. oxygen consumption rate discrepancy
- the discussion is too long, and repetitive with redundant affirmations; authors should insteand focus on conflicting metabolic data, clinical relevance of reduced IL-6/IL-8, and why your findings deviate from other studies reporting positive effects of red/NIR light on proliferation
- preliminary nature and methodological limitations should be highlighted in the introduction section too
Reviewer 3 Report
Comments and Suggestions for Authors This report relates to a study on isolated stromal cells from adipose tissue of volunteers. The ‘clinical application’ (lines 101-103) would involve isolation of these cells and irradiation, with the treated cells then ‘re-transferred to the same patient’. This report proposes that treatment of a minor fraction of total adipose tissue could somehow have a ‘regenerative’ result. What volume of tissue and cells might be involved? Would it be a significant portion of the total number of stromal cells present? It is proposed that blue light can have adverse effects (line 617). Fortunately, it penetrates very poorly into tissues. Effects of red light might promote ‘cell growth’ but anyhow do not ‘impair’ (line 627). What need to be explained: [1] what volume of ‘treated’ cells will be generated, and [2] could it have any significant effect ‘in vivo’?
The introductory material claims that longer wavelengths of light can penetrate into tissues ‘up to 5.4 mm’. Blue light will penetrate to ever shorter distances. Line 99 discusses cell suspensions. A procedure whereby stromal cells were ‘freshly isolated from a patient’ is mentioned, but no protocol is described for obvious reasons. From where would these cells be obtained, how would they be converted to a cell suspension and how transferred back?
Cells were isolated from volunteers by liposuction and treated with assorted wavelengths of light. These suspensions were then exposed to different wavelengths of light and combinations of these. There appears to be no description of the procedure used to assess ‘viability’ after treatment. Something involving acridine orange and DAPI was involved. The DCFDA assay will detect reactive oxygen species, but these will immediately interact with any suitable molecule in the vicinity, so use of DCFDA after irradiation detects only long-persisting lipid peroxides. The correct procedure involves adding DCFH before irradiation.
What is ‘metabolic activity’ (section 2.7)? This assay measures the activity of some mitochondrial enzymes. While data on one group of enzymes can be assessed, the claim that this an accurate measure of total metabolic activity is unsubstantiated. What is termed ‘adipogenic differentiation’ is assumed to be related to lipid accumulation.
Results: total ROS formation can only be determined if the probe is present during irradiation. Fig. 2 indicates that some long-persisting species are formed by 455 nm light. Use of 455 nm light appears to reduce the cell number (Fig. 4). There appears to be no significant effect of light on ‘metabolic activity’, whatever this might mean. This is also true for other parameters that are assessed. MTS may be ‘commonly used’ (line 513) but it does not evaluate ‘viability’ which can only be assessed by clonogenic assays or similar direct tests for cell division.
Summary: it is proposed (lines 652-654) that a surgical procedure involving isolation and irradiation of adipocytes could be used to promote wound healing. Since this scenario is to take place ‘in the operating theatre’ (line 654), it is clearly not ‘non-invasive’ (line 657). Since only a very small volume of cells will be treated, it is not clear how a significant effect is anticipated.
Round 2
Reviewer 2 Report
Comments and Suggestions for Authors
The authors have answered all my comments. The work is ready for publication.
Author Response
Thank you for your valuable comments, which have helped us to eliminate inaccuracies and improve the focus of our manuscript.

Reviewer 3 Report
Comments and Suggestions for Authors
In the prior review, issues were raised which the authors have mainly ignored. The authors continue to refer to ‘specific wavelengths of light’: all wavelengths are ‘specific’. What is the basis for the ‘viability and cell count’ assay (section 2.4)? The unambiguous method for assessing viability is a clonogenic assay. As pointed out before, the DCFH-DA assay should be done correctly with the reagent present during irradiation. Otherwise, only a few long-persisting ROS will be detected. The correct procedure and rationale for its use have been amply described in the literature. It is those ‘transient reactive oxygen species’ (line 223) that are responsible for biologic effects. Oleinick has shown (Photochem Photobiol 2009) that MTS and similar assays do not reliably indicate effects of treatment on the ability of cells to proliferate. Why is proliferation important? Is it expected that treated adipocytes are going to proliferate?
In the section beginning with line 74, it is proposed that this report is associated with ‘regenerative medicine’. The authors mention ‘wound healing’, ‘angiogenesis, ‘immune responses’ and ‘tissue repair’. All of this is supposed to be accomplished by irradiating adMSCs. In the Results section, it is claimed that treatment of these cells has effects on cell number, ‘metabolic activity’, inflammatory responses, migration ability, wound closure and adipogenic differentiation.
What ‘regenerative therapies’ might be developed from this procedure? The protocol involves isolation of adMSCs by liposuction follower by a digestion and filtration procedure. CD34-positive cells are isolated and irradiated. In lines 451-453, it is proposed that a variety of beneficial effects might be produced, presumably by transfusing these cells back to patients. There is no evidence presented that any of these effects occur in vivo.
Assuming that irradiation can produce ROS, these will be cytotoxic to cells. If the DCFH procedure was done correctly, there might have been a substantial level of ROS formed upon irradiation if the appropriate wavelengths were used. The authors note (lines 486-488) that ROS formation can reduce both viability and therapeutic efficacy. What is the point of irradiation?
How ROS formation will enhance ‘cellular resilience’ (line 496) is not explained.
After a discussion of various effects of red vs. blue light, the authors propose that use of red or near IR light might be useful. It is not clear what volume of adipocytes would be processed to create a sufficient number of cells for an in vivo effect. Data consistent with such an effect is shown in Fig. 7 where migration of adMSCs was claimed to be enhanced by irradiation. There is substantial overlap of data shown in panel b) with perhaps a hint of efficacy. But these data were obtained using a pure population of treated adMSCs. In an in vivo situation, the number of treated cells will represent only a small portion of the total number of such cells present.
What needs to be shown: [1] What beneficial effects are produced upon irradiation of adMSCs? [2] What volume of treated adMSCs will be needed to create a significant beneficial effect in vivo? [3] What protocol would be used?
Round 3
Reviewer 3 Report
Comments and Suggestions for Authors Experimental conditions were modified such that phenol red and FBS were removed during exposure of cells to light. Whole lack of serum can have metabolic effects on cells, the duration should be sufficiently short to minimize any effects. It would be helpful if light doses were indicated in terms of mJ/sq cm rather than in duration, e.g., in Table 2. What is the basis for the determination of ‘viability’ using the assay described in line 203? The DCFH assay was carried out with the reagent added after irradiation. Since ROS have a very limited half-life, this will not accurately report on ROS formation and will identify only the long-lived species, e.g., lipid peroxides. Many investigators have carried out studies where the reagent is present during irradiation. While the authors to correctly indicate what is being detected, this assay will not report on the much more pertinent reactive oxygen species being formed. While the authors do indicate that their determination occurred ‘after light exposure’, most of the ROS being formed were not detected. What is ‘metabolic’ activity (section 2.8)? Oleinick carried out a study in 2009 (Photochem Photobiol) and found that such studies did not predict for clonogenic survival of cells. Metabolic activity is a meaningless statement. What is an ‘energy-dependent reduction in cell number’ (line 354). Section 4.2 contains material that should be modified. It is indicated that ’immediately after light exposure’ there were no detectable effects. This is not unexpected. Effects of irradiation on cell division will not be apparent until there has been sufficient time for cells to divide (or not). What is ‘prolonged PDT’ (line 505)? What does ‘the prolonged PDT’ mean (Line 515)? How would exposure to blue light affect the immune response (line 563)? In the legend to Table A1, there should be an indication of how ‘viability’ was assessed. This can only be unambiguously determined by clonogenic assays. Line 342 does not describe the method. The ‘population doubling time’ measurement could mean that either some cells were unable to proliferate after treatment, or that the doubling time of the entire cell population was slowed by irradiation.Author Response
Please see the attachment
